# African Swine Fever Outbreak in an Enclosed Wild Boar Hunting Ground in Serbia

**DOI:** 10.3390/pathogens12050691

**Published:** 2023-05-09

**Authors:** Jasna Prodanov-Radulović, Jovan Mirčeta, Biljana Djurdjević, Sava Lazić, Sanja Aleksić-Kovačević, Jelena Petrović, Vladimir Polaček

**Affiliations:** 1Scientific Veterinary Institute “Novi Sad”, 21000 Novi Sad, Serbia; 2Vojvodina Šume Enterprise, 21000 Novi Sad, Serbia; 3Faculty of Veterinary Medicine, University of Belgrade, 11000 Belgrade, Serbia

**Keywords:** ASF, enclosed hunting ground, Serbia, wild boars

## Abstract

African swine fever (ASF) has been detected in many European countries since its introduction in Georgia in 2007. Serbia suffered its first case of ASF in the domestic pig population in 2019. At the beginning of 2020, ASF was detected in wild boars in open hunting grounds in the southeastern region of the country in districts along the country’s borders with Romania and Bulgaria. Since then, all ASF outbreaks in wild boar were clustered in the population located in the same bordering areas. Despite the newly implemented biosecurity protocols for hunters in 2019, ASF was detected for the first time in June 2021 in the wild boar population located in an enclosed hunting ground in the northeast region of the country. In this study, we reported the first ASF outbreak in a wild boar population located in an enclosed hunting ground in close proximity to the Serbian–Romanian border. The epizootiological data on the field investigation of the ASF outbreak, with descriptions of the clinical signs and gross pathological lesions detected, including the total number as well as the estimated age, sex, and postmortem interval, were analyzed. Clinical signs were detected only in nine diseased wild boars, while in total, 149 carcasses were found in the open and enclosed part of the hunting ground. In addition, 99 carcasses from which samples (parts of spleen or long bones) were collected for molecular diagnostics (RT-PCR) were confirmed as ASF-positive. The results of the epidemiological investigations indicate the central role of wild boar movements as well as the constant risk of human-related activities in the countries bordering area.

## 1. Introduction

African swine fever (ASF) is a contagious viral disease in hosts belonging to the *Suide* family, caused by a double-stranded DNA virus, a member of the family *Asfarviridae* and the genus *Asfivirus* [1,2,3]. Since the ASF virus (ASFV) was first introduced in Georgia (2007), it has progressively spread on the European continent [4,5,6]. The current ASF epizootic in Europe affects both domestic pigs and wild boars [7,8,9]. According to available data, the majority of ASF notifications in Europe were made in wild boars, suggesting that this population represents the predominant host of ASF [8,9,10]. In wild boar populations, the onset of ASF was characterized by an initial epizootic wave with almost 100% lethality [6,11], followed by a long-term endemic status [5,12], which makes the control of the disease extremely difficult [13,14,15]. After entering its endemic phase, ASF seems to be maintained essentially by infected carcasses, which act as a virus reservoir [5,7,9]. According to Pepin et al. [16], ASF can persist with a low prevalence through three main mechanisms: live infected wild boars, convalescent but still infectious individuals, and the high tenacity of the virus in carcasses and the environment. The specific epidemiology in the wild boar habitat cycle, as well as the drivers of persistence and transmission routes, is still not fully understood [4,6,12,15]. The first case of ASF in Serbia was confirmed in 2019 in the domestic pigs in a village located in the central region of the country [17], several hundreds of kilometers away from the nearest reported ASF outbreak, suggesting a human-mediated disease introduction over long distances [4,12]. In early 2020, ASF was detected for the first time in wild boars in the southeastern region of the country, in districts close to the border with Romania and Bulgaria [18,19]. Since then, numerous outbreaks have been reported in wild boar in open hunting grounds close to the border area in southeastern Serbia [18,19,20]. However, in June 2021, ASF was detected for the first time in the wild boar population located in the northeastern region of the country (Vojvodina Province), in the border area with Romania. At the beginning of 2019, surveillance activities and newly implemented biosecurity protocols for hunters were introduced in the country bordering area. 

In this study, we report the first ASF outbreak in a wild boar population in an enclosed hunting ground and, thus, the first disease detection in the northeastern region of the country, close to the Serbian–Romanian border. 

## 2. Materials and Methods

### 2.1. Description of Wild Boar Hunting Grounds

The research was conducted in the wild boar hunting grounds “Vršačke planine” (altitude from 200 to 641 m) in Vojvodina Province, Serbia. The investigated hunting area (4118.38 hectares) includes an open and enclosed fenced part (301.67 ha). The total number of wild boars was 253, including 73 in the open hunting area and 180 in the fenced area. The fenced part of the hunting ground is secured with a single wire mesh fence (height 2.20 m), with wooden posts at 3 m. The wire mesh fence is interspersed with natural water surfaces (streams, flash floods). Protective metal grates have been placed at the stream entrance to prevent other wildlife from entering while allowing the free flow of water. The hunting ground is surrounded by other open hunting grounds managed by local hunting clubs. The eastern side of the fenced part of the hunting ground is only 500 m from the Serbian–Romanian border (Figure 1). 

### 2.2. Epizootiological Data, Postmortem Assessment, Sample Collection, and Laboratory Analysis

The data used for this study are descriptive epizootiological data obtained during the outbreak field investigations. The epizootiological data included the date, geographical coordinates, estimated age, and time since death of found carcasses. After the initial report, a detailed epizootiological investigation was conducted that included filed inspections, interviews with forest workers, necropsy, and sampling for laboratory testing. 

Pathological alterations were evaluated and scored according to the guidelines regarding macroscopic changes in experimentally ASF-infected wild boars [23]. Tissue samples (part of spleen) or long bones were collected in order of ASFV genome detection [24] at the National ASF Reference Laboratory (NRL). For the ASFV genome detection, real-time PCR was carried out as a 25 µL reaction using primers described by King et al. (2003) [24]. The molecular assays were performed in the NRL according to the protocols described by Glisic et al. [25]. 

Field data were analyzed using ESRI Arc GIS Desktop software, ESRI ArcMap 10., ESRI ArcGIS Online, ESRI ArcGIS Pro 3. and Microsoft Excel 365.

After official confirmation of the ASF, the veterinary authorities ordered measures in accordance with the national legislation. 

## 3. Results

### 3.1. Epizootiological Investigation in Open Part of Hunting Ground 

The first wild boar carcass was reported by the game warden on 14 June 2021 (male, estimated age 6 months and postmortem interval (PMI) up to 24 h). Sampling was conducted as part of the official passive surveillance program. Within less than 24 h, the presence of the ASFV genome was confirmed by molecular testing in the NRL. The second carcass was found the next day in the vicinity of the first one (male, estimated age 14 months, and PMI up to 24 h). Three days later, two wild boar carcasses (male, age 6 months and sow 3 years old) were found in an advanced (6–8 day) stage of decomposition near the water surface, 300–400 m from the fence of the fenced portion of the hunting ground. Two days later, two more wild boar carcasses in an advanced decomposition stage (6–8 day) were found in new locations (two sows, estimated age 3 years). In the open part of the hunting ground, a total of only six carcasses were found (three juvenile males, 6–14 months, and three sows). However, according to a report from the hunting ground manager, after three weeks of active terrain search, it was estimated that the initial wild boar population (73 wild boars in total) had decreased by about 50%. In open hunting areas, wild boar population density is regularly monitored and assessed by game wardens according to the Official Guidance on Methods for Wild Boar Abundance and Density Estimation [26]. According to the official report, wild boar density estimation was enhanced by observing the feedlots, resting sites, natural water sources, wild boar tracks, and pellet counts. In addition, the movement of wild boar into neighboring hunting grounds was not reported.

### 3.2. Epizootiological Investigation in the Enclosed Hunting Ground

On 23 June 2021, the first wild boar carcass was found (estimated aged 2 years and PMI up to 7 days), followed by two juvenile and one adult female carcass (PMI 1–4 days). Over the next three days, the next five carcasses were found in close proximity to the stream (estimated PMI more than 7 days). Thereafter, in total, 60 wild boar carcasses were found exclusively in the stream, but later on, they were found in all locations in the fenced part of the hunting ground. Finally, there were no traces close to the feedlot and other areas that indicate the presence of wild boars. On 30 August 2021, the last carcass was found. According to hunting ground data, the total number of wild boars before the ASF outbreak was 180, but only 143 carcasses were found. The remaining 37 wild boars were not found by the end of the active area search. The assumption of the hunting manager was that sick wild boars died and the carcasses were hidden in the forest vegetation or eaten by the local scavengers. The hunting ground manager estimated the mortality rate in the fenced part of the hunting grounds to be 90–100%. In total, 99 carcasses from which samples were collected for molecular diagnostics (RT-PCR) were confirmed as ASF-positive: 6 carcasses were found in the open, and 93 were found in the enclosed hunting ground. The samples for testing included the part of the spleen (for carcasses with estimated PMI 1–3 days (13 in total), while in the remaining carcasses, the test sample was a long bone (86 in total) (Appendix A). Finally, the Serbian Veterinary Authority made the decision that the remaining 44 found carcasses were not going to be tested. They were safely collected and buried in the previously designated part of the hunting ground. The estimated age distribution of the carcasses found in both parts of the hunting grounds was: 35 juvenile wild boars and 114 adults (73 sows and 41 wild boars). The locations of the dead wild boars found in the hunting ground “Vršačke planine” are shown in Figure 2. 

The hunting ground was characterized by rocky terrain, which does not allow for burying the carcasses. In the described ASF outbreak, the burying of wild boar carcasses was approved by a special provision of Serbian law. The burial pit for all carcasses was dug in the open part of the hunting grounds, in a location where the ground was 4 m deep and not underwater. 

### 3.3. Clinical and Gross Pathological Findings

Clinical signs (slower movement when running, incoordination, foot paddling) were observed only in nine diseased wild boars in the fenced part of the hunting ground. Due to the presence of decomposing carcasses during the summer season, only 9 of the 21 wild boards were partially evaluated postmortem. Gross findings of all examined animals revealed changes in the skin, lymphoid tissue, spleen, and kidneys. The external postmortem evaluation of nine wild boars revealed erythematous to hemorrhagic areas of skin, mostly visible in hairless zones (score 1). Furthermore, on internal examination, hemorrhagic lymphadenopathy was observed in all nine cases on mandibular, mesenteric, and gastro-hepatic lymph nodes. These were all dark red and enlarged (score 3). Epicardial and endocardial petechiae were present in three cases (Score 1). In all animals, the spleen was enlarged and dark red congested (score 3). In two cases, mild to moderate petechial hemorrhages were also noted on intestinal serosa as well (score 2). In addition, cortical petechial hemorrhages (mild to moderate) were detected in the kidneys (score 1).

## 4. Discussion

In this study, we reported the first ASF outbreak in a wild boar population in an enclosed hunting ground near the Serbian–Romanian border. In diseased animals, notified clinical signs and postmortem lesions strongly suggest an acute course of ASF [3,11], with almost 100% lethality. In neighboring Romania, a total of 1081 outbreaks in wild boars and 1637 ASF outbreaks in domestic pigs were reported from September 2020 to August 2021 [21,27,28]. Nevertheless, in two administrative districts (Timis and Caras-Severin) located on the border with Serbia, 7 outbreaks in domestic pigs and 29 ASF outbreaks in wild boars were officially reported in the period from January to June 2021 [22,29]. Since there are no fences and/or natural barriers restricting the movement of wild boars in the border region, it can be assumed that animals regularly cross the border between countries. In the current epizootic wave, ASF is frequently introduced by the cross-border movement of wild boars to new territories (e.g., Baltic States, from Poland to Germany) [10,30,31]. Several hypotheses have been considered for the introduction of ASF into the enclosed hunting grounds: human-mediated, flash flooding and seasonal wild boar movement in the adjacent country bordering region. 

It is well known that human activities in the forest (hunting, logging, wild boar supplemental feeding, mushroom and berry picking, etc.) may play a role in the indirect transmission of ASF [15,21,32]. In current ASF epizootics, humans have often been considered the main cause of both long-distance transmission and the virus’s introduction into uninfected wild boar habitats [7,9,15,30]. In described Serbia–Romania border area, intensive cross-border human activities, and travel occur daily. In addition, there are intensive family relationships with relatives on both sides of the border, who have certain privileges when it comes to frequent border crossings. Especially in spring and summertime, short-time touristic visits of people from the bordering areas of Romania and the city of Timisoara are frequent.

The existing fence between the open and enclosed parts of the hunting ground is interspersed with natural water bodies: streams and flash floods that flow unhindered through the fences, i.e., from the open to the fenced part of the hunting ground. Protective metal grates were placed at the stream entrance to prevent the entry of large game and other wildlife while allowing the free flow of water. Considering the geographical features of the hunting ground (hill-mountain type), it could be assumed that previously reported local flash floods [33] could carry away the carcass parts or their fragments and even penetrate the metal grid between the fenced and open part of the hunting ground. Indeed, the location of the first carcass in the open part of the hunting ground coincided with the course of the flash flood with the location of the first carcass in the fenced area. Finally, both parts of the hunting ground are surrounded by open hunting areas where the seasonal movement of wild boar from Romania is a common phenomenon. It is generally accepted that wild boars show strong side fidelity, and most stay within 1–2 km of their home ranges [34]. The individuals may disperse relatively short distances on a daily basis (less than 5 km), but longer dispersals have also been observed [10]. However, the wild boars are settled when there is a sufficient food supply, and they can move about 10–20 km per day because of human disturbances (e.g., four-wheel motor vehicles) [35].

Longer movements are characteristic of wild boar males. Indeed, during the breeding season, the mobility of the males increases in search of receptive females. However, the movements of young wild boars leaving their maternal groups are considered the main source of long-distance movements in populations [10]. It is assumed that the average herd of wild boar moves about 1–2 km in the wintertime and up to 10–12 km during the summer [35]. A number of similar studies have shown that the movement of wild boars poses the highest risk of ASF introduction and spread [8,10,13,14].

Since no vaccine against ASF is available [3,4] in the European continent recommended control strategy in the wild boar population nowadays includes fencing the affected area, removal of carcasses, and targeted hunting [4,5,21]. Certainly, all of the above measures need to be supported and strengthened by the implementation of biosecurity measures during hunting to minimize the likelihood of human-mediated ASF spread and prevent its entry into the pork production system [7,8,18].

## 5. Conclusions

It may be concluded that wild boars’ seasonal movements and human-related activities in the bordering area probably played the most important role in ASF transmission. The seasonal wild boar movements are mostly related to the males in the breeding season (October–November), but also wild boar herd movement toward the feed source, i.e., corn fields (June–August). In the meantime, human-related intensive agricultural activities in the fields, as well as in the forest (forest harvesting with heavy machines), can be viewed as artificial factors that may be the reason for unseasonal wild boar movements in the bordering area.

The presented results of the ASF outbreak in an enclosed hunting ground suggest that additional control measures need to focus on the borders of the country and that joint efforts with neighboring countries could be a step forward to ASF stop transmission. One of the joined efforts might be fencing in order to prevent the migration of potentially infected wild boar across national borders. Future research will certainly need to focus also on the physical characteristics of the conceptual fence (i.e., the specific fence depth in the ground, a vegetation-free zone near the fence) and the specific imprint of the human population living in the border region itself. In addition to estimation of the animal density and monitoring activities, small-scale landscape fragmentation needs to be considered a factor that may reduce disease transmission in Serbia–Romania bordering areas.

## Figures and Tables

**Figure 1 pathogens-12-00691-f001:**
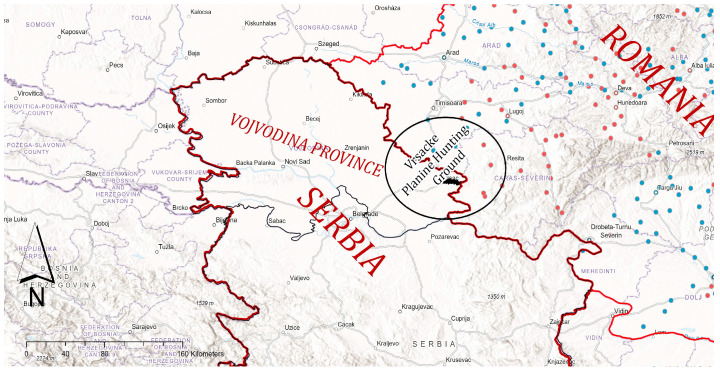
Geographical location of the hunting ground “Vršačke planine” in Vojvodina Province. Legend: red dots represent confirmed ASF outbreaks in domestic pigs in Romania; blue dots represent confirmed outbreaks in wild boars in Romania [21,22].

**Figure 2 pathogens-12-00691-f002:**
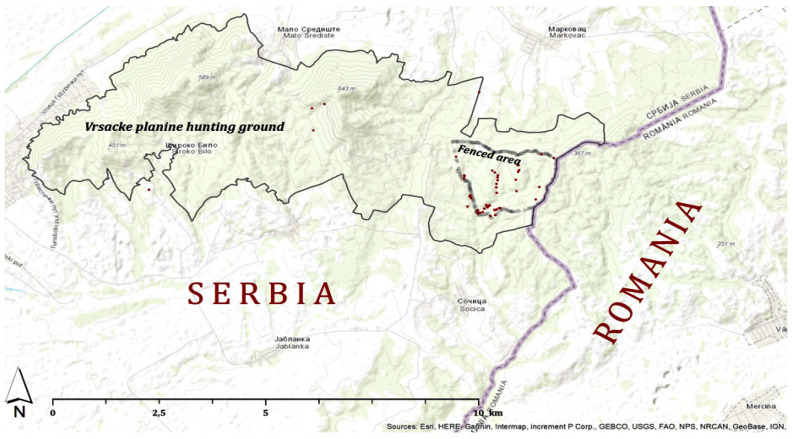
The location of the open and fenced parts of the hunting ground. Legend: red dots represent the geographic location of the wild boar carcasses that tested positive for ASF.

## Data Availability

Not applicable.

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
