# Peer review of "African Swine Fever Outbreak in an Enclosed Wild Boar Hunting Ground in Serbia"

_pathogens, 2023, doi:10.3390/pathogens12050691_

Round 1

Reviewer 1 Report (New Reviewer)

This study provides a descriptive epidemiological analysis of ASF in wild boars reported from hunting areas in the northeastern region of Serbia, with mention of possible introduction of infection from neighboring countries. Some minor modifications are noted below.

L33-35: Some may wonder why endemics occur despite a 100% mortality rate. I suggest adding a sentence explaining this.

L96-98: What does the dramatic decline in the population mean? Has it moved to another area because it is an open area, or are there other possible factors? Please state the authors' thoughts.

L150-154: As the authors say, anthropogenic factors are one of the most important factors in carrying the virus to distant locations. Is there frequent movement of people and goods to and from infected areas in Romania in the study area? It would be good to insert a sentence mentioning this.

L164-165: The authors may add the reference.

L174: It is recommended that the authors be more specific about the meaning of seasonal movement of wild boars in this sentence. For example, do the authors believe that the movement of male wild boars around during the breeding season can lead to the spread of infection? It would be good to state any findings regarding the association between the timing of infection reports and the timing of their seasonal movements.

Author Response

Reviewer 2 Report (New Reviewer)

ASF is an economically important infectious disease for global pig industry. Up to date, the commercial vaccine is still unavailable, and any treatment is forbidden, therefore the only way to control the disease is provided by strict sanitary measures and and hunting wild boars to reduce their population in order to limit the spread of the disease. The communication entitled: “African Swine Fever Outbreak In An Enclosed Wild Boar Hunting Ground in Serbia” described the major role wild boar movements as well as constant risk of human-related activities in the countries border as one of the causes of the spread of ASF.

 In my opinion, the communication requires revision and language correction

Line 16: it is known that wild boars will be concentrated in the wild boar population, no need to repeat the word “wild boars”

Line 16,19,139, 142: In the case of wild boar, the nomenclature has been standardized and should be used “outbreaks”

Line 29-30, 33, 145, 152: instead of the word “epidemic” there should be “epizootic”, epidemic refers to humans and epizootic refers to animals.

Line 26-37: The word ‘wild boars” was used 5 times in 5 sentences , the Authors should use synonyms

Line 31-32: “were made in wild boars, indicating that the wild boar population” this is purple prose

Line 43-46: Both sentences describe the same situation twice.

Lie 46-48: To keep the chronology, the authors should put this sentence as the second in this paragraph

Line 70-71: Authors should use the word “outbreaks” instead of “cases”

Line 96-98: On what basis was the 50% decrease in the wild boar population estimated?

Line 108: What happened to the other 37 wild boars since it was a closed hunting area? why don't the authors mention this in the text?

Line 110-115: Shouldn't the found wild boars carcasses be disposed of in a dedicated facility? and not buried in the ground still being a potential source of ASF infection?

Line 118-119: The authors should specify that they are talking about the carcasses of wild boars that died as a result of ASF

Paragraph 3.4 where did the 9 diseased ASF wild boars come from, which the Authors did not mention earlier in the text, on the basis of observations of which the authors described clinical symptoms of slower movement when running, incoordination, foot paddling

Line 144: Overuse of the phrase "wild boars" in one sentence, the authors could have replaced it with, for example, the word: "animals"

Line 155-158: The sentence is written very vaguely, it can be inferred from it that the wild boars carcasses or their fragments can get outside the fence

Line 163-165: The data given by the Authors are inaccurate, how can you understand the statement: "It is generally accepted that a wild boar's area of movement is 5 km under normal conditions. Under the conditions of seasonal movement, this distance increases to 30 km." ? wild boars cover this distance in a day a month, a year or per hour?

Paragraph 5.0: Germany's fencing at the Polish border did not work because ASF was confirmed in Germany in 2022. The authors should develop the thought and expand the conclusions, because fencing alone is not enough

 This communication needs minor corrections and clarification of certain aspects. Towards the end of the report there is information about 9 diseased wild boars that were subjected to both vital and post-mortem observation but nowhere is it described what these boars were, when they were observed, only the clinical signs "(slower movement when running, incoordination, foot paddling)" are mentioned. The authors should pay closer attention to the misuse of the phrase "wild boars" even several times in one sentence, they should use synonyms. The nomenclature of confirmed outbreaks of disease among animals has been standardized, and these are outbreaks both in wild boars and in pigs.

Author Response

Reviewer 3 Report (New Reviewer)

African Swine Fever is rapidly spreading across different parts of Europe and Asia and poses a significant threat to pigs and pork industry (pork producers and people working in the industry).  ASF is perhaps one of the most dangerous viral diseases, with its high mortality rate, transmission and lack of effective treatments. It spreads rapidly through contact with infected animals or contact with contaminated pens, trucks, clothing or feed. Pigs can also remain carriers for the disease for quite some time. Since ASF is challenge for pig health worldwide, both passive and active surveillance in both, domestic pig and wild boar populations, is very important. This paper gives fresh information on ASF situation in Eastern part of Europe. However, I do suggest certain things, which need attention, improvement and clarification to support and strengthen the overall impact of the article.

Points for attention:

There are no methods, results or conclusions in the abstract.

What type and how many tested samples must be indicated in the abstract.

The aim of the research was not indicated in the Introduction section.

It is necessary to provide data about the number of carcass samples collected, storing conditions, methods used for the DNA isolation (also storing conditions of the isolates, how the quality of isolates was monitored…), PCR (were positive/negative controls used), sequencing in the Materials and Methods section. Since the authors declare that the molecular testing was conducted in NRL, authors didn’t write sufficient data about protocols used.

There is no data regarding sequencing. Was the sequence analysis performed? If it was, were the sequences similar to those isolated in neighboring country (Romania)?

Sub sections 3.1. and 3.2. are more suitable for the Materials and Methods section than for Results.

Round 2

Reviewer 3 Report (New Reviewer)

Dear authors, thank You for the revision of Your manuscript and for all the explanations.

This manuscript is a resubmission of an earlier submission. The following is a list of the peer review reports and author responses from that submission.

Round 1

Reviewer 1 Report

The authors in this proposed manuscript do not present any clear scientific question  , they do not present any information about the data analysis of the epidemiological data in the appropriate section. It looks as they took an internal report and tried to fit it into a peer-review scientific manuscript.

Reviewer 2 Report

African swine fever outbreak in an enclosed wild boar hunting ground in Serbia

By Radulovic et al, 

The paper is interesting, well written, and describes the epidemiological situation that occurred in an area on the border between Serbia and Romania. 

The following are my comments:

a. It would be important to add a comment regarding the Romanian situation just across the border; i.e. whether the presence of the virus is known and whether in wild boar or pigs (or both). Such news would make the epidemiological picture clearer and thus also make it easier to carry out sub-chapter 4.2, which at present does not allow any conclusions.   

b. The figures and in particular No. 2 and No. 3 should be enlarged as it is currently not possible to fully understand the details. 

Line 37: introduced instead of transferred

Line 46: replace JUMPS with transmission; 

Line 128 (2.1.): specify that this is a single net fence; describe type of supply, seasonality, quantity and how many times a week or month it is supplied

Line 138 specify whether Red or Roe DEER  

Lines 150-151: difficult to understand. Explain what is meant by productive area etc. 

Line 154: not clear COMPARED TO? 

Line 178: VERY? What does it mean in this context? 

Line 197: unclear; what does (CONCRETE) IRON mean? To be rewritten. 

Line 270: replace tubular bone by LONG BONE here and wherever the diction tubular bone is present. 

Sub-chapetr 3.3. I would suggest adding a table highlighting the number of deaths inside and outside the fence and their periods. Sometimes it is difficult to follow the long description of the cases; adding a table would make it easier to read. Figure 2 should also distinguish between cases inside and outside the fence (no distinction is currently visible and the figure is far too small).   

Line 360: typo in REFERENCE;

Line 370: replace MIGRATION by Movements or Seasonal movements. The boar does not make true migrations. Change MIGRATION everywhere in the text. 

Chapter 4: The epidemiological link with the situation in Romania is definitely missing here. 

Line 461: Difficult to understand what CORRELATION BETWEEN HUNTED/FOUND DEAD WAS 6,42. What epidemiological significance does this division have? With what data (no. hunted where? When?) has it been calculated? This part should be better explained. 

Line 570: What is meant by CALVING PERIOD?    Do you consider this form to be correct? Please kindly check. 

Line 573: A bibliographic entry should be added to support the statement that young females return to heat after losing their offspring. It is generally believed that young females go into heat when they reach the appropriate weight (around 30-35 kg), which happens later in the year. 

Line 576: Here too, the assertion of the presence of POLYESTRICITY in wild boar and domestic pig-boar hybrids should be supported by at least one scientific citation. 

Line 642: remove EAGLE (Buzzard is the correct name not Buzzard eagle);

Lines 677-695. Here there seems to me to be some confusion between the role of boar movements (not migrations) and spread of infection. The main mechanism of virus transmission does not occur through the movement of infected wild boar, but rather through the continued spatial distribution of the species that forms and maintains a continuous chain of infection. This part of the work must be reorganised by considering the main mode of transmission, i.e. that by continuity of spatial distribution of the wild boar.   

Reviewer 3 Report

This scientific paper describes the ASF situation in Serbia, focused on the wild boar population. The work is interesting but the structure of the manuscript is still a work-in-process. Authors should focus on their data and describe the findings in a more accurate and concise way. Missing scorings (pathomorphological findings for example) and lack of statistics make the manuscript very weak. Finally, an extensive English proofreading is also a must.

Introduction

Introduction should be shortened and more accurately explained. Authors repeat some terms and hypothesis too often during this part of the paper. A more precise and concise introduction to the topic will allow the reader to have a better understanding.

Additional information about the possible epidemiology between domestic pigs and wild boar outbreaks can also be discussed using the information given by the German reference laboratory about the spread of the ASFV variants, that could be found in clusters and isolated from both neighborhood wild boar populations and outbreaks in domestic pig farms.

L14: in „a“village

L18: at the country's borders

L21: “we report the first ASF case”

L 37-38: please, rephrase this statement, it is not grammatically correct

L49: was

L 57-58: please rephrase this sentence, they are not grammatically correct

Material and methods

Wild boar hunting ground description

This part should focus on the wild boar and the area, information about breeding and hunting of other ungulates is here irrelevant for the scope of the paper. ASF is a suid disease and other ungulates, such as roe deer or fallow deer are not infected. For this reason, this part of the paper should be shortened and explained more accurately to describe the carried-out work.

Epidemiological data

Please add information about data management and curation. Did you use a specific software to recover or analyze the data?

Results

Results should be better structured and more accurately exposed. The amount of information given in the text misleads the reader about the outbreak and the clinical/pathological signs found in the carcasses. I suggest to restructure the results and depict them using graphical approaches (i.e., GIS).

Pathomorphological findings should also be more accurately described and scored using international guidelines (i.e. https://www.mdpi.com/2076-0817/9/9/688 ).

L 243: 'Carcass' is a dead body of an animal, whereas 'corpse' is a dead body, especially of a person.

Figure 2 does not present enough quality

Discussion

This part of the paper should also be shortened and more concise. Discussion should focus on the obtained results and its comparison with related countries and outbreaks. I suggest to rewrite this part of the manuscript.
